# Integrated Methylome and Transcriptome Analysis Provides Insights into the DNA Methylation Underlying the Mechanism of Cytoplasmic Male Sterility in Kenaf (*Hibiscus cannabinus* L.)

**DOI:** 10.3390/ijms23126864

**Published:** 2022-06-20

**Authors:** Zengqiang Li, Dengjie Luo, Meiqiong Tang, Shan Cao, Jiao Pan, Wenxian Zhang, Yali Hu, Jiao Yue, Zhen Huang, Ru Li, Peng Chen

**Affiliations:** 1Key Laboratory of Plant Genetics and Breeding, College of Agriculture, Guangxi University, Nanning 530004, China; lizengqiang2020@163.com (Z.L.); luodengjie01@126.com (D.L.); tangmeiqiong2006@163.com (M.T.); caoshan202276@163.com (S.C.); panning@126.com (J.P.); wenxzhang1314@163.com (W.Z.); yalihh@163.com (Y.H.); yuejiao1120@163.com (J.Y.); hz549935163@gmail.com (Z.H.); 2Henan Collaborative Innovation Center of Modern Biological Breeding, Henan Institute of Science and Technology, Xinxiang 453003, China; 3College of Life Science & Technology, Guangxi University, Nanning 530004, China; liruonly@163.com

**Keywords:** kenaf, cytoplasmic male sterility (CMS), anther and pollen, DNA methylation, transcriptome

## Abstract

Cytoplasmic male sterility (CMS) is widely exploited in hybrid seed production. Kenaf is an important fiber crop with high heterosis. The molecular mechanism of kenaf CMS remains unclear, particularly in terms of DNA methylation. Here, using the anthers of a kenaf CMS line (P3A) and its maintainer line (P3B), comparative physiological, DNA methylation, and transcriptome analyses were performed. The results showed that P3A had considerably lower levels of IAA, ABA, photosynthetic products and ATP contents than P3B. DNA methylome analysis revealed 650 differentially methylated genes (DMGs) with 313 up- and 337 down methylated, and transcriptome analysis revealed 1788 differentially expressed genes (DEGs) with 558 up- and 1230 downregulated genes in P3A compared with P3B. Moreover, 45 genes were characterized as both DEGs and DMGs, including *AUX,*
*CYP*, *BGL3B*, *SUS6*, *AGL30* and *MYB21*. Many DEGs may be regulated by related DMGs based on methylome and transcriptome studies. These DEGs were involved in carbon metabolism, plant hormone signal transduction, the TCA cycle and the MAPK signaling pathway and were shown to be important for CMS in kenaf. These results provide new insights into the epigenetic mechanism of CMS in kenaf and other crops.

## 1. Introduction

Kenaf (*Hibiscus cannabinus* L.) is an annual fiber crop belonging to the *Malvaceae* family with great development potential; it is widely used in the textile, papermaking, building and feed industries. Kenaf displays fast growth, high yield and strong tolerance to abiotic stress, such as salt and heavy metal stress. [1,2,3].

Plant cytoplasmic male sterility (CMS) is a maternally inherited pollen sterility trait useful for controlled crosses that exploit heterosis. Our research group initially discovered a kenaf cytoplasmic male sterility mutant (UG93A) and bred the first kenaf hybrid cultivar (Hongyou 1) using this CMS line [4]. It is important to understand the molecular mechanism underlying CMS to better produce novel kenaf CMS lines and utilize heterosis. We previously carried out comparative transcriptome, proteome and posttranslational proteome analyses in the kenaf CMS line and its maintainer line, and the results showed that the identified differentially expressed genes, differentially accumulated proteins and differentially modified proteins participated in various biological processes, such as carbohydrate and energy metabolism, gene expression and signal transduction, thus playing important roles in kenaf CMS [1,5,6,7]. These works provide a comprehensive and systematic basis for further understanding the molecular mechanism of CMS in kenaf. However, more efforts are still needed to understand this complicated CMS trait in depth.

Carbohydrates provide nutrition and energy for plant growth and development, especially pollen development. Furthermore, they also act as signals in the regulation of pollen development and interact with phytohormones [8,9]. In recent years, an increasing number of studies have shown that the lack of sugars and ATP may be one of the main factors causing plant male sterility. Liu et al. reported that carbohydrate and energy metabolism in CMS *Brassica napus* were significantly downregulated [10]. Oliver et al. reported that sucrose imbalance in anthers can affect pollen development, and ABA is a potential signal for CIPS (cold-induced pollen sterility) [11]. Zhang et al. reported that reduced MST8 (monosaccharide transporters 8) expression in rice csa (carbon starved anther) mutant plants resulted in a lack of sugar supply, which ultimately led to male sterility [12]. In addition, many studies have detected lower ATP production in some CMS flowers [13,14].

DNA methylation refers to the addition of a methyl group to a cytosine. It mainly occurs within CpG and non-CpG cytosines in all kinds of eukaryotic cells. In plants, DNA methylation is one of the most extensively studied epigenetic modifications [15]. It has an important function in many biological processes, including seed germination, cellular differentiation, vernalization, plant flowering, gene expression, and plant growth and development [16,17]. It has been reported that DNA methylation in different plants is related to morphological development, abiotic stress response, and agronomic traits [16,17,18]. Several studies have reported associations between DNA methylation and CMS in plants. In rice, the DNA methylation pattern correlates with CMS occurrence [19], and DNA methylation may be involved in the regulation of fertility-related gene expression in the sterile rice line PA64S [20]. DNA methylation levels in fertility-restored hybrids were higher than those of a sterile maize line [21]. In soybean, many genes altered by DNA methylation are involved in carbohydrate and energy metabolism, transcriptional regulation, male gametophyte growth, and pollen and flower development, and these genes are closely related to CMS [22,23]. In cotton, many differentially methylated genes are also involved in carbohydrate metabolism and male gametophyte development [15]. However, there is no information available on the relationship between DNA methylation and the CMS mechanism in kenaf.

In the present study, comparative physiological and DNA methylation combined with transcriptome gene expression analyses in the kenaf CMS line P3A and its maintainer line P3B were conducted. The results provide new insights into the mechanisms of kenaf CMS from the point of view of DNA methylation-regulated gene expression.

## 2. Results

### 2.1. Hormones, Photosynthetic Products and ATP Contents

Plant hormones adjust to all aspects of plant growth and development [24,25]. Carbohydrates and ATP are the main energetic components and materials supplied to plant cells; they are often correlated with genetically determined growth and even the occurrence of CMS [1]. Therefore, the contents of plant hormones, including indoleacetic acid (IAA), gibberellic acid (GA), abscisic acid (ABA), photosynthetic products (including starch, soluble sugar and sucrose) and ATP in the anthers of kenaf CMS and maintainer lines were investigated and compared. As a result, compared with the maintainer line P3B, the IAA and ABA contents in the CMS line P3A were significantly lower by 14.56% and 26.48%, respectively, the GA contents were significantly higher by 15.56% (Figure 1a), and the contents of starch, soluble sugar, sucrose and ATP were significantly lower by 23.66%, 33.66%, 33.99% and 43.71%, respectively (Figure 1b,c).

### 2.2. Genomic DNA Methylation Levels in Kenaf

The results of data filtering and methylation site coverage are shown in Table 1. A total of 18.28 and 17.51 million raw reads were generated in the kenaf CMS line P3A and maintainer line P3B, respectively. After filtering, the methyl RAD sequencing data (enzyme reads) included 5.26 and 5.73 million reads for P3A and P3B, respectively. Finally, 4.72 and 5.25 million mapping reads were generated. All enzyme reads were submitted to the Sequence Read Archive (SRA, https://www.ncbi.nlm.nih.gov/sra/; accessed on 29 September 2019) of NCBI (accession numbers: SRR10163906, SRR10163907). A total of 120,323 CCGG and 63,240 CCWGG DNA methylation sites were characterized in P3A, with average methylation coverages of 3.51 and 4.05, respectively. A total of 131,529 CCGG and 65,376 CCWGG DNA methylation sites were found in P3B, with average methylation coverages of 4.03 and 4.17, respectively. These results show that the total DNA methylation level of P3A was lower than that of P3B.

### 2.3. Distribution of Methylated Sites in Different Functional Elements and Gene Regions

The methylated sites upstream (within 2 kb upstream of the gene promoter), downstream (within 2 kb downstream of the gene terminator), exons, 1st exons, introns, genes, intergenic regions, splice site acceptors, splice site donors, splice site regions, 5′ terminal untranslated regions (5′ UTRs) and 3′ terminal untranslated regions (3′ UTRs) were determined (Figure 2). The results showed that the methylation sites were mainly located in intergenic regions, followed by downstream, upstream, intron and exon regions, but there were fewer methylation sites in other functional elements (5′ UTR, splice site region and 3′ UTR). The distribution trends of CCGG sites were consistent with those of CCWGG sites. On each functional element, the distribution of methylation sites in P3A was less than that in P3B, especially in CCGG sites.

The distribution of methylation sites 2 kb upstream and downstream of the transcription start site (TSS), 2 kb upstream and downstream of the transcription termination site (TTS) and in gene-coding regions was analyzed. The results (Figure 3 and Appendix A) showed that the distribution of methylation sites in the gene-coding region was significantly higher than that in the upstream and downstream 2 kb regions of the TSS and TTS, and the methylation level of P3A in the gene-coding region was lower than that of P3B. These results imply that the gene-coding region of kenaf had a higher DNA methylation level, and the DNA methylation level in the male sterile line P3A was lower than that in its maintainer line P3B, which may be one of the reasons for CMS.

### 2.4. Differentially Methylated Sites (DMSs) and Differentially Methylated Genes (DMGs) between P3A and P3B

According to the sequencing depth information of the methylation sites, the methylation levels of each gene were calculated, and the differentially methylated genes (DMGs) of the CMS line P3A and maintainer line P3B were compared. Using IsoSeq full-length cDNA sequences, a total of 2806 unique methylated sites were identified as differentially methylated sites (DMSs), collectively containing 1792 CCGG sites and 1014 CCWGG sites. These 2806 unique methylated sites corresponded to 650 DMGs, including 521 DMGs with CCGG sites and 129 DMGs with CCWGG sites. Among those CCGG sites in DMGs, 274 and 247 were identified as up- and downregulated, respectively; for the CCWGG sites in DMGs, 39 and 90 were characterized as up- and downregulated, respectively (P3A vs. P3B, *p* < 0.05, log_2_FC > 1, Table 2 and Appendix A).

### 2.5. GO and KEGG Enrichment Analysis of DMGs

GO enrichment analysis showed that 79 DMGs were highly enriched in 68 GO terms. The total enriched GO terms are shown in Appendix A, and the top ten GO terms in each category are shown in Figure 4. In the biological process category, DMGs were mostly enriched in regulation of transcription, auxin-activated signaling pathway, methylation, negative regulation of flower development and ethylene-activated signaling pathway. In the cellular component category, integral component of membrane, cytoplasm, endoplasmic reticulum, and mitochondrion were highly enriched. In the molecular function category, DMGs were highly enriched in ATP binding, DNA binding and peroxidase activity.

KEGG pathway enrichment analysis showed 41 highly enriched DMGs that were distributed among 56 metabolic pathways (Appendix A). The top 20 enriched pathways are shown in Figure 5. DMGs were involved in pathways such as starch and sucrose metabolism, plant hormone signal transduction, carbon metabolism, protein processing in endoplasmic reticulum, photosynthesis, glycolysis/gluconeogenesis, RNA polymerase, galactose metabolism, inositol phosphate metabolism, AMPK-signaling pathway and fructose and mannose metabolism, which play vital roles in the regulation of plant growth and development. Combined with the results of previous physiological indicators, this study showed that DNA methylation, especially DMGs, was involved in the regulation of carbon metabolism and plant hormone signal transduction.

### 2.6. Differentially Expressed Gene (DEG) Analysis

After filtering the low-quality raw reads, transcriptome sequencing produced 42.30 and 37.37 million paired-end clean reads with Q30 values of 96.25% and 96.36% from P3A and P3B, respectively. The complete clean reads for these libraries in this study were deposited in the Sequence Read Archive (SRA) (https://www.ncbi.nlm.nih.gov/sra/; accessed on 30 August 2019) of NCBI with the accession numbers SRR10039125 and SRR10039126. These reads were aligned to the kenaf reference genome, and the alignment rates of P3A and P3B were 86.94% and 91.92%, respectively.

A total of 1788 genes were differentially expressed in P3A, including 558 (31.21%) up- and 1230 (68.79%) downregulated genes (Appendix A). This result indicates that most of the DEGs showed a decreasing expression tendency in the CMS line P3A compared with the maintainer line P3B.

### 2.7. GO and KEGG Enrichment Analysis of DEGs

The results showed that 1233 DEGs were enriched in 820 GO terms, and each term included two or more DEGs. The GO terms carbohydrate metabolic process, auxin-activated signaling pathway, pollen tube growth, pollen germination, cytoplasm, chloroplast, integral component of membrane, ATP binding, DNA-binding transcription factor activity and glucan endo-1,3-beta-D-glucosidase activity were highly enriched (Figure 6 and Appendix A), and these GO terms were closely related to anther and pollen development [5,6,7].

The results of the KEGG pathway enrichment analysis showed that 405 DEGs were enriched in 149 KEGG metabolic pathways. KEGG metabolic pathways such as starch and sucrose metabolism, carbon metabolism, plant hormone signal transduction, glycolysis/gluconeogenesis, biosynthesis of amino acids, carbon fixation in photosynthetic organisms, galactose metabolism and MAPK signaling pathway were highly enriched (Figure 7 and Appendix A). Most DEGs were downregulated, especially in carbon metabolism and plant hormone signal transduction. These results indicated that many DEGs disturbed plant hormone signal transduction and carbon metabolism and decreased the synthesis of IAA, ABA, carbohydrate and ATP in the male sterile line P3A. This decreased synthesis cannot meet the needs of pollen development and may be one of the main reasons for CMS.

### 2.8. Integrated Analysis of DMGs and DEGs

A total of 45 genes were characterized as both DEGs and DMGs. To analyze the relationship between DNA methylation and gene expression levels, a four-quadrant graph was drawn to show DNA methylation and gene expression levels (Figure 8, Appendix A). Out of those 45 genes, 8, 13, 10 and 14 genes were located in the first, second, third, and fourth quadrants, respectively. The DNA methylation of genes located in the first and third quadrants may positively regulate gene expression, while that of genes located in the second and fourth quadrants may negatively regulate gene expression. The genes encoding chalcone synthase (*CHS2)*, proline dehydrogenase (*proDH)*, *bHLH 39*, alpha-crystallin domain-containing protein (*alpha22.3*) and RNA polymerase beta subunit (*rpoB*), located in the first quadrant, were both upregulated as DMGs and DEGs. In the second quadrant, the genes encoding RNA-binding protein 38 (*RBP38*), beta-amyrin synthase (*β-amyrin*), cytochrome P450 (*CYP*), terpene synthase (*TPS9*), serine/threonine-protein phosphatase (*PP 7*), WAT1-related protein (*WAT1*), KN motif and ankyrin repeat domain-containing protein (*Kank3*) were downregulated as DMGs but upregulated as DEGs. The genes encoding acid beta-fructofuranosidase-like (*βF*), beta-glucosidase (*BGL3B*), fasciclin-like protein (*FLA12*), fatty acid amide hydrolase (*FAAH*), alpha-amylase 2 (*mal2*), LIM domain-containing protein WLIM2b (*WLIM2b*) and organic cation/carnitine transporter 4 (*OCTN4*), located in the third quadrant, were both downregulated as DMGs and DEGs. In the fourth quadrant, the genes encoding LRR receptor-like serine/threonine-protein kinase (*LRR-RLK*), histone-lysine N-methyltransferase (*HKMT*), protein WEAK CHLOROPLAST MOVEMENT UNDER BLUE LIGHT 1 (WCMUBL1) (*WEB1*), and xylan alpha-glucuronosyltransferase (*xagt4*) were upregulated as DMGs but downregulated as DEGs. Most of these genes were reported to be involved in plant growth and development. Several genes with uncharacterized functions were found to be both DMGs and DEGs. This suggests that DNA methylation regulates gene expression in different patterns. However, we cannot completely attribute differential expression to DNA methylation, as there are other factors that may also regulate expression. Further study is needed to explore the relevance of these DMGs and DEGs.

### 2.9. qRT–PCR Verification

To confirm the reliability of transcriptome sequencing data and reveal the correlation between DNA methylation and gene expression, we randomly tested 27 key genes involved in various biological processes, such as DNA methylation, hormone signaling, transcription factors, flower development, the TCA cycle and energy metabolism. The expression patterns of these genes were highly consistent with the results of transcriptome sequencing. At the same time, the results showed that DNA methylation regulates gene expression in different manners (Table 3).

## 3. Discussion

DNA methylation can cause changes in chromatin structure, DNA conformation and the interaction between DNA and protein, thus regulating gene expression [26]. However, the molecular mechanisms by which DNA methylation controls the expression of gene networks involved in pollen development and CMS occurrence in kenaf remain largely unclear. In the present study, the contents of IAA, ABA, starch, soluble sugar, sucrose and ATP were significantly decreased in CMS line P3A compared with its maintainer line P3B (Figure 1a–c); 650 DMGs (Appendix A) and 1788 DEGs (Appendix A) were identified, and among them, 45 genes (Appendix A) were characterized both as DEGs and DMGs. According to our integrated DNA methylome and transcriptome analysis, many DEGs and DMGs could be used for the detection of CMS occurrence and pollen developmental processes.

### 3.1. DMGs and DEGs Related to Carbohydrate Metabolism Processes and the TCA Cycle

Carbohydrates or sugars are essential to fundamental plant growth processes [8]. Starch and sucrose are the main products of photosynthesis in higher plants. Starch is a key substance in plant metabolism, and even small disturbances in starch turnover may affect metabolism and growth [27]. Sucrose is an important factor in fruit quality and cell metabolism and participates in gene expression regulation [28]. Starch synthase and sucrose synthase play important roles in regulating starch and sucrose, respectively, and sucrose synthase also participates in the regulation of starch synthesis. The TCA cycle is the main source of energy for life activities; it produces more ATP than other respiratory pathways, and it is also the hub of material metabolism in plants. Studies have shown that the demand for ATP during pollen development is significantly increased, and insufficient ATP supply may be one of the main reasons for pollen abortion [1]. In the present study, many DMGs and DEGs were related to carbohydrate metabolic processes and TCA pathways (Appendix A), including malate dehydrogenase (*MDH*), starch synthase (*SS*), sucrose synthase (*SUS*), glucose-6-phosphate (*G6P*), citrate synthase (*CS*), pyruvate kinase (*PYK*), and UDP-glucose 6-dehydrogenase (*UGDH*). As an example, the methylation level of sucrose synthase *SUS6* in P3A was 8.86 times higher than that in P3B, and its expression level was only 6.8% of that of P3B. It is speculated that increased DNA methylation of this gene results in a downregulation in the CMS line. As a result, the contents of starch, soluble sugar, sucrose and ATP in anthers in P3A were significantly lower than those in P3B (Figure 1b,c). This implies that DEGs and DMGs related to carbohydrate metabolic processes and the TCA cycle can play a role in kenaf CMS.

### 3.2. DMGs and DEGs Related to Plant Hormone Signal Transduction

Plant hormones play an important role in regulating the growth and development of plants. Gibberellin can induce flowering in plants [29]. Auxin not only promotes cell division and plant growth but also regulates nutrient transport and distribution and promotes female flower differentiation and plant flowering [24]. Interruption in auxin biosynthesis, transport or signaling can lead to flowering defects [25]. In the present study, the IAA content in the CMS line P3A was significantly lower than that in the maintainer line P3B; however, the GA content was significantly higher in the CMS line (Figure 1a). It is speculated that the significant changes in IAA and GA contents were closely related to the occurrence of CMS in kenaf. Many DMGs and DEGs, including auxin transporter-like protein (*LAX*), auxin-induced proteins (*AUX*), indole-3-acetic acid-amido synthetase (*GH*) and SAUR-like auxin-responsive protein family members (*SAUR*), were found. Therefore, it could be speculated that the change in the methylation level influences the content of phytohormones and pollen abortion in the CMS line.

### 3.3. DMGs and DEGs Related to Cytoskeletal Organization

The cytoskeleton is mainly composed of microtubules, microfilaments and intermediate filaments, which maintain the morphology and internal structure of the cytoplasm and promote the communication of material and information. A weakened cytoskeleton in microsporocytes has been reported to cause male sterility in many species of plants [30,31,32]. In wheat, pollen abortion is related to actin content [33]. Actin depolymerizing factor (ADF) plays an important role in gametophyte formation, pollen tube elongation, and plant growth and development. Mutations in ADFs from different species have been associated with lethality, arrest in cell proliferation, pollen germination and pollen tube growth [34,35]. Yan et al. reported that the actin gene was mainly expressed in pollen, and its expression level in male sterile plants was much lower than that in its maintainer line. Moreover, they further obtained male sterile wheat and tomato plants by antisense expression of the actin gene [33]. The present study demonstrated that 18 DEGs were enriched in the regulation of the actin cytoskeleton, and the genes encoding actin, actin-depolymerizing factor (*ADF*) and actin-related protein 2/3 complex subunit 5A-like (*ARPC5A*) were strongly downregulated in the CMS line. Thus, we speculated that the downregulation of the expression of these genes may cause cytoskeletal organization impairment in pollen, eventually leading to CMS in kenaf.

### 3.4. DMGs and DEGs Related to Transcription Factors (TFs) 

Transcription factors are closely related to plant development and responses to the environment [36]. During the development of male gametes in *Arabidopsis thaliana*, more than 600 transcription factors interact with each other to form a dynamic regulatory network [37]. In the present study, many TFs were identified as DMGs or DEGs in the CMS line. Some of these TFs may play crucial roles in the normal development of gametes and floral organs in kenaf.

Among transcription factors, the MYB family is one of the largest TF families in plants. These TFs are widely involved in the regulation of plant development and metabolism and play an important regulatory role in anther and pollen development. Studies have shown that *AtMYB21* and *AtMYB24* regulate the development of stamens through interactions with jasmonate (JA) [38,39]. In a previous study, pollen and anther development of *Arabidopsis AtMYB21* and *AtMYB24* mutants was defective, eventually leading to male sterility [40]. *AtMYB26* regulates anther dehiscence by affecting secondary thickening of the endothecium; for this reason, the anthers of the *AtMYB24* mutant could not dehisce and release pollen, thus resulting in sterility [41]. In this study, the expression level of *MYB21* in the kenaf CMS line P3A was increased by 14 times, while the expression level of *MYB26* was only half that of the maintainer line P3B. It is speculated that these significant changes in gene expression levels play a role in the occurrence of kenaf CMS. MADS-box genes play key roles in regulating the development of floral organ differentiation. A subset of pollen-specific MIKC-type MADS-box proteins (*AGL30*, *AGL65*, *AGL66*, *AGL94*, and *AGL104*) are expressed preferentially during pollen maturation [42]. Double and triple mutants of the *AGL65*, *AGL66* and *AGL104* genes showed decreased pollen activity and inhibited pollen tube germination [43]. In the present study, 31 MADS-box genes were differentially expressed, and 10 of them were differentially methylated, including *AGL30*, *AGL66* and *AGL104*. Therefore, it could be speculated that MADS-box genes may also play crucial roles in the normal development of pollen in kenaf.

In addition to MADS boxes and MYBs, other TFs belonging to the WRKY, bHLH and NAC families were also characterized as DMGs or DEGs in the present study. All of the above-described DMGs and DEGs may play a role in flower development and fertility in kenaf.

### 3.5. DMGs and DEGs Involved in MAPK and Calcium-Dependent Signalling Pathway

Mitogen-activated protein kinases (MAPKs) are serine/threonine protein kinases that respond to various growth factors in cells and are involved in plant stress resistance, hormone signal transduction, cell cycle regulation and gamete development [44]. Calcium-dependent protein kinases (CDPKs) are the most important calcium-sensing proteins and play a key role in plant calcium signal transduction. As CDPKs are involved early in calcium signal transduction, they can regulate a large number of processes [45]. In this study, 17 transcripts involved in the MAPK signaling pathway and 45 transcripts involved in the calcium-dependent signaling pathway were found to be downregulated in the CMS line P3A, and three transcripts involved in the calcium-dependent signaling pathway were also identified as DMGs. For example, RAS-related proteins are mainly involved in the activation of MAPK signaling [46]. Phospholipase C is an important regulatory enzyme in calcium-dependent signaling pathways and participates in ABA signal transmission [47]. Moreover, there have been many reports on the regulation of pollen development by phospholipase C [48,49].

## 4. Materials and Methods

### 4.1. Plant Materials

The kenaf cytoplasmic male sterile (CMS) line P3A and maintainer line P3B were used in this study. Both P3A and P3B were grown in an experimental field under field conditions and normal management. The anthers from both lines were collected at the dual-core period (pollen abortion stage of the CMS line). Anthers from three different plants of each line were harvested, pooled, quickly frozen in liquid nitrogen and stored at −80 °C for further analysis.

### 4.2. Determination of Physiological Indexes

The contents of indoleacetic acid (IAA), gibberellic acid (GA) and abscisic acid (ABA) in anthers were determined by using a double-antibody sandwich ELISA kit with three biological repetitions (Jiangsu Jingmei Biotechnology Co., Ltd., Taixing, China; IAA: JM-01121P1, GA: JM-110047P1, ABA: FK2876; and BCA kit for protein quantification, article number JM-100009A). The contents of total soluble sugar, starch, sucrose and ATP in anthers were measured separately by using a determination kit with three biological repetitions (Shanghai Solarbio Bioscience & Technology Co., Ltd., Shanghai, China; article numbers: BC0030, BC0070, BC2460 and BC0300, respectively).

### 4.3. MethylRAD Library Construction, Sequencing, and Data Analysis

The anther DNA of the CMS line P3A and maintainer line P3B was extracted by a plant genomic DNA kit (Cat No. Dp35-03, TIANGEN, Beijing, China), and the quality and concentration of DNA were determined by 1.0% agarose gel electrophoresis and a NanoDrop2000 spectrophotometer. A MethylRAD-seq sequencing library was constructed using the method described by Wang et al. [24]. Briefly, the endonuclease *FspE*I (NEB, Ipswich, MA, USA), which recognizes C^m^C sites (C5 methylation and C5 hydroxymethylation), was selected for genomic DNA digestion. The digested DNA was then connected with two adaptors with cohesive ends by T4 DNA ligase (NEB, USA). *FspE*I was cut at the 12th and 16th recognition sites of the 3’ end and viscous end, so the joint was a 4 bp NNNN. The fragments with adaptors were enriched by PCR amplification with specific primers (first-round amplification, primers were designed according to the adapter sequence). The products were subjected to 8.0% polyacrylamide gel electrophoresis, and fragments of approximately 100 bp in length were recovered and purified. The purified fragments were subjected to the second round of PCR amplification (primers were designed according to the adapter and sequencing connector). After 1.0% agarose gel electrophoresis, the target bands (30 bp) were recovered and purified, and MethylRAD sequencing was performed using the Illumina HiSeq X^TM^ Ten platform. MethylRAD sequencing and analysis were conducted by OE Biotech Co., Ltd. (Shanghai, China).

The sequencing data were filtered to remove sequences containing adaptors, N-bases and low-quality sequences, as well as sequences without restriction sites, and finally, high-quality sequencing data containing methylation sites (MethylRAD label) were obtained. The sequences containing *FspE*I recognition sites (C^m^CGG and C^m^CWGG) were subsequently aligned against the reference genome of kenaf using Bowtie 2 (version 2.3.4.1) with default parameters. Bedtools (V2.25.0) was used to calculate the distribution of methylation sites in different gene elements of each sample. RPM (reads per million) values were used to quantify the methylation levels of each methylated site, and the methylation degree of genes was calculated as the sum of sequencing depths of all methylation sites in a gene. Differential *p*-values (*p* < 0.05) and fold changes (Log_2_FC > 1) of each site (gene) between groups were analyzed by the R package DESeq [50], and GO and KEGG annotation was performed for differentially methylated genes.

### 4.4. cDNA Library Preparation, Sequencing and Data Analysis

Total RNA was extracted from anthers of P3A and P3B by a mirVana Rotal RNA Isolation Kit (Invitrogen, Thermo Fisher Scientific Inc., Waltham, MA, USA). Total RNA quality and purity were determined by a NanoDrop ND-1000 spectrophotometer (NanoDrop), and RNA integrity was detected by an Agilent 2100 Bioanalyzer (Agilent Technologies, Santa Clara, CA, USA). The libraries were constructed using the TruSeq Stranded mRNA LT Sample Prep Kit (Illumina, San Diego, CA, USA) according to the manufacturer’s instructions. Sequencing was performed by the HiSeq X^TM^ Ten platform by OE Biotech Co., Ltd. (Shanghai, China). Clean reads were obtained by filtering the raw reads for adaptor, poly-N and low-quality sequences. Clean reads were compared to the kenaf reference genome using HISAT2 software [51]. Gene expression was calculated by the FPKM (fragments per kb per million reads) method using Cufflinks software [52,53]. The R language package Deseq2 was used to screen differentially expressed genes (DEGs). The screening criteria were a *p*-value < 0.05 and |log_2_FoldChange| ≥ 1. To investigate the biological functions of DEGs, Gene Ontology (GO) and Kyoto Encyclopedia of Genes and Genomes (KEGG) annotations were analyzed by Blast2GO and KOBAS2.0, respectively [54,55].

### 4.5. qRT–PCR Analysis

Total RNA of the kenaf CMS line P3A and its maintainer line P3B was reverse transcribed into cDNA using a reverse transcription kit (Vazyme Biotech Co., Ltd., Nanjing, China). qRT–PCR was performed using ChamQ Universal SYBR qPCR Master Mix (Vazyme Biotech Co., Ltd., China) on a Bio-Rad CFX96 (Bio-Rad Laboratories). The reaction mix was 10 μL qPCR mix, 4 μM forward primers and reverse primers, 10 ng cDNA template and 20 μL ddH_2_O. The reaction procedure was as follows: 95 °C for 30 s, followed by 40 cycles at 95 °C for 10 s and 60 °C for 30 s. *Histone 3* (*H3*) was used as an endogenous reference gene [56], and the relative expression of genes was calculated by the 2^−ΔΔCT^ method. The primer sequences for qRT–PCR are shown in Appendix A.

### 4.6. Statistical Analysis

Excel 2016 and SPSS 22 were used for physiological data processing and statistical analysis; Excel 2013 and GraphPad Prism 7 were used for chart drawing. All data were analyzed by one-way ANOVA (ANOVA, SPSS 23.0), with *p* < 0.05 indicating significant differences. The results are expressed as the mean ± SD.

## 5. Conclusions

Comprehensive analysis of DMGs and DMGs based on GO and KEGG pathway analyses concluded that CMS in P3A might be caused by altered DNA methylation and disturbed gene expression. The DEGs and DMGs involved in the carbohydrate metabolic process and TCA cycle, plant hormone signal transduction, cytoskeleton organization, transcription factors and the MAPK and calcium-dependent signaling pathway may play profound roles in kenaf CMS (Figure 9). The results provide new thoughts and insights into the epigenetic mechanism of kenaf CMS and pollen development, as well as a basis for understanding this complicated mechanism in other species. Based on the results of this study, we will further explore the functions of both DEGs and DMGs that are closely related to the occurrence of CMS, such as *AUX*, *SUS6*, *AGL30*, *MYB21* and RAS-related proteins, and the role of DNA methylation in their gene expression.

## Figures and Tables

**Figure 1 ijms-23-06864-f001:**
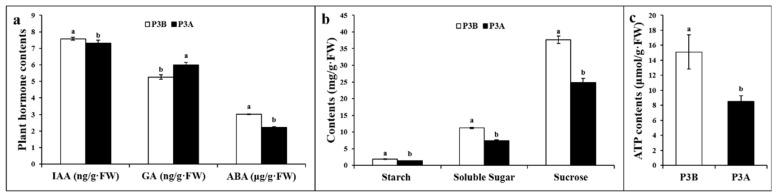
Comparison of the plant hormone, photosynthetic product, and ATP contents. (**a**) IAA, GA and ABA contents; (**b**) starch, soluble sugar and sucrose contents; and (**c**) ATP contents. Note: The different lowercase letters indicate significant differences at the *p* < 0.05 level.

**Figure 2 ijms-23-06864-f002:**
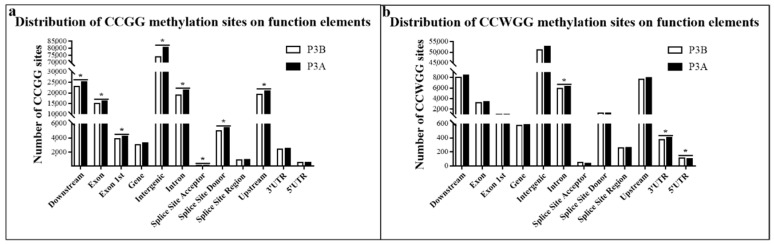
Distribution of methylation sites in different functional elements. (**a**) CCGG sites; (**b**) CCWGG sites. Note: The asterisk indicates significant differences at the *p* < 0.05 level.

**Figure 3 ijms-23-06864-f003:**
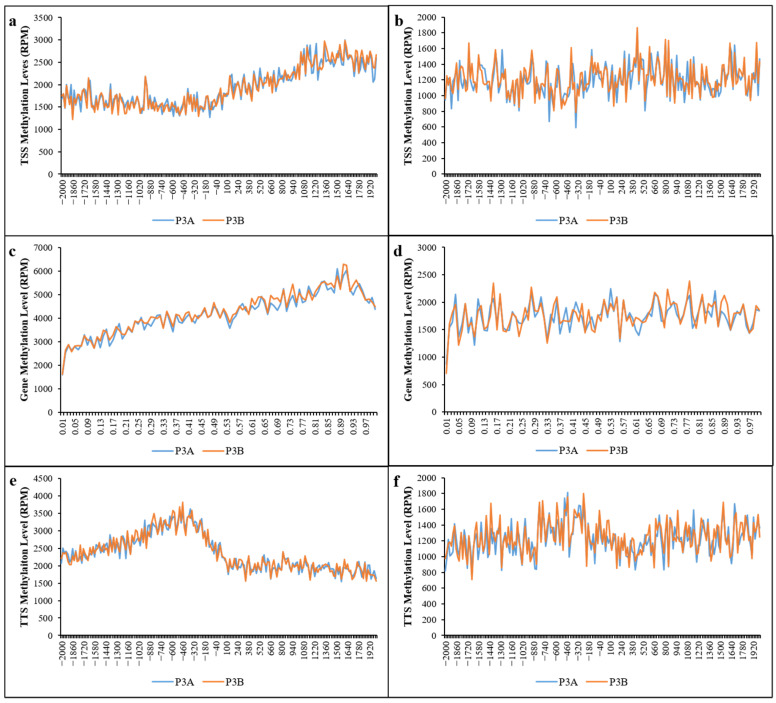
Distribution of methylation sites in the gene body. (**a**,**b**) Distribution of CCGG and CCWGG methylation sites 2 kb upstream and downstream of the transcription start site (TSS) region, respectively; (**c**,**d**) distribution of CCGG and CCWGG methylation sites in the gene-coding region, respectively; (**e**,**f**) distribution of CCGG and CCWGG methylation sites 2 kb upstream and downstream of the transcription termination site (TTS) region, respectively. Note: (**a**,**b**,**e**,**f**), the transverse axis represents the location of the methylation region at the TSS (or TTS) of genes, where a negative value indicates the number of bases upstream and a positive value indicates the number of bases downstream. (**c**,**d**), the transverse axis represents the degree of methylation, expressed as 0–1.

**Figure 4 ijms-23-06864-f004:**
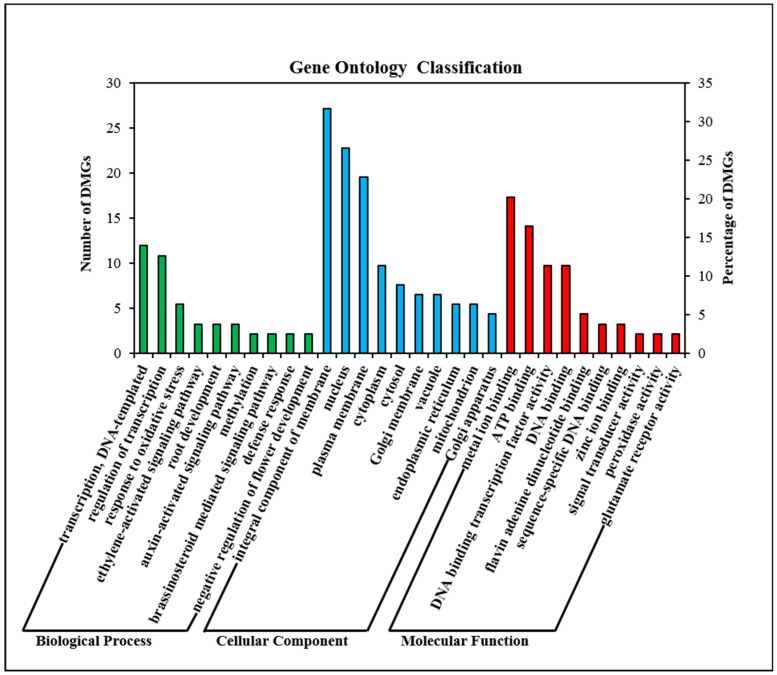
GO enrichment analysis of differentially methylated genes (DMGs).

**Figure 5 ijms-23-06864-f005:**
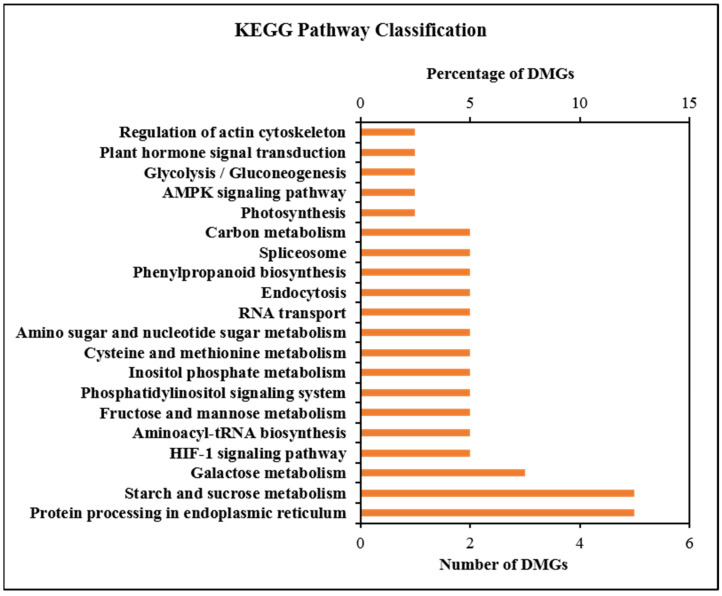
KEGG enrichment analysis of differentially methylated genes (DMGs).

**Figure 6 ijms-23-06864-f006:**
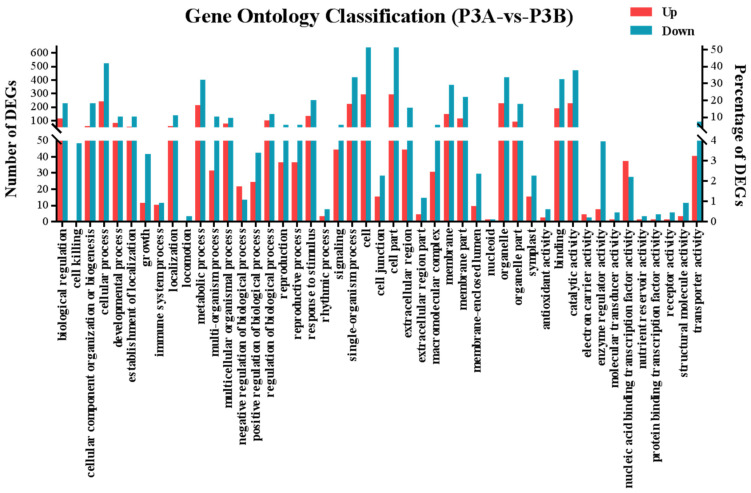
GO enrichment analysis of differentially expressed genes (DEGs).

**Figure 7 ijms-23-06864-f007:**
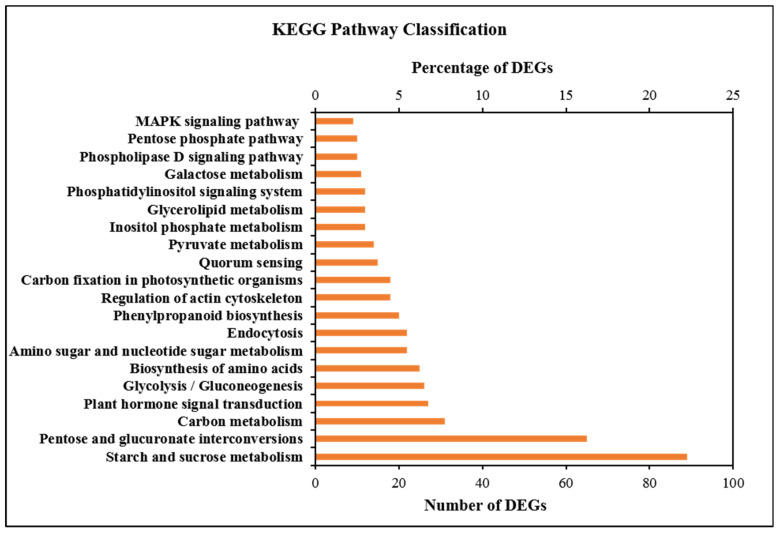
KEGG enrichment analysis of differentially expressed genes (DEGs).

**Figure 8 ijms-23-06864-f008:**
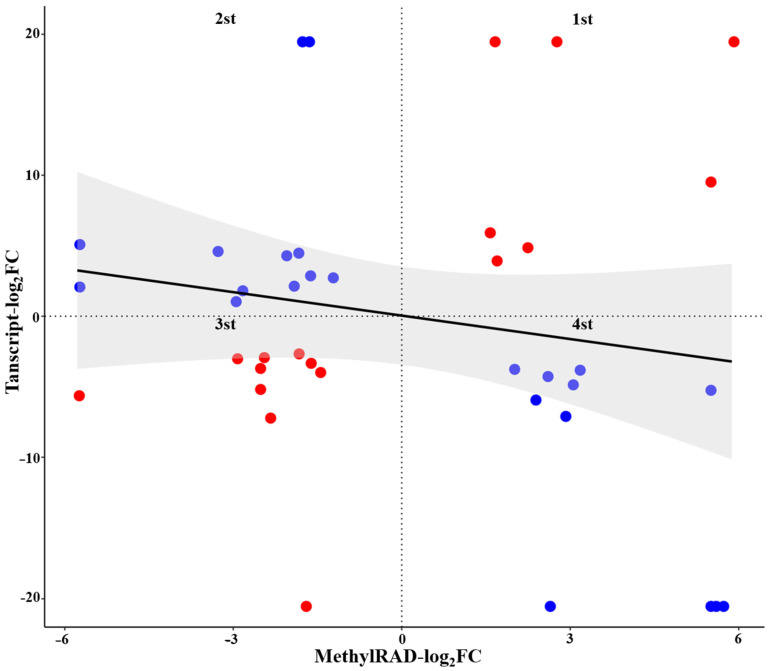
The correlation between differentially expressed genes (DEGs) and differentially methylated genes (DMGs). Note: Red dots indicate that the methylation level of the gene is negatively correlated with its expression level, and blue dots indicate that the methylation level of the gene is positively correlated with its expression level. The grey shadow indicates that the correlation between the methylation level and expression level is not significant (*p* > 0.05).

**Figure 9 ijms-23-06864-f009:**
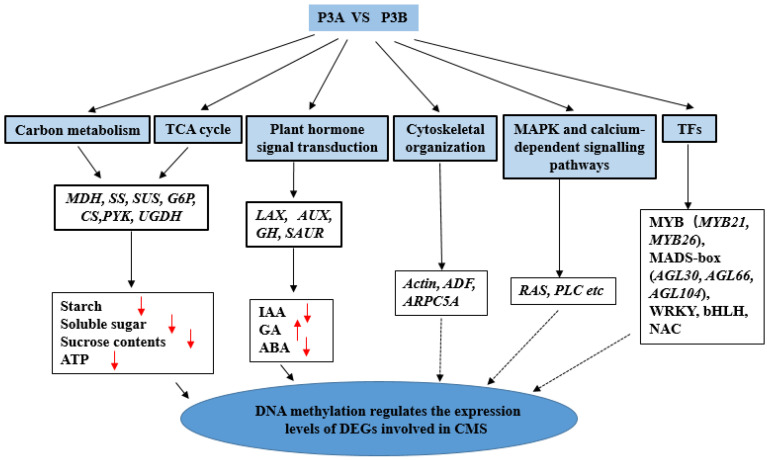
A model for the changes in DMGs and DEGs contributing to CMS in kenaf. Note: The upwards red arrowhead indicates that the content of this indicator is significantly increased in P3A, and the downwards red arrowhead indicates that the content of this indicator is significantly decreased in P3A. The solid line indicates that there has been sufficient and in-depth research, while the dotted line indicates that further in-depth research and verification are needed.

**Table 1 ijms-23-06864-t001:** Summary of sequencing quantity and methylation site coverage.

Sample	Raw Reads	Enzyme Reads	Mapping Reads	Mapping Reads Ratio	CCGG Sites		CCWGG Sites	
					Number	Depth	Number	Depth
P3A	18,283,466	5,264,564	4,716,458	89.59%	120,323	3.51	63,240	4.05
P3B	17,513,339	5,733,725	5,246,070	91.50%	131,529	4.03	65,376	4.17

**Table 2 ijms-23-06864-t002:** The number of DMSs and DMGs.

Diff_Type	DMS Number		DMG Number	
	CCGG Site	CCWGG Site	CCGG Site	CCWGG Site
Diff	1792	1014	521	129
Up	786	446	274	39
Down	1006	568	247	90

**Table 3 ijms-23-06864-t003:** qRT–PCR validation of the RNA-seq data.

Gene Name	Functional Annotation	Fold Change (P3A vs. P3B)		
		Methylation	Transcriptome	qRT–PCR
*atp8*	Phospholipid-transporting atpase 8	2.18	0.49	0.47
*COX2*	Cytochrome c oxidase subunit II	2.19	4.53	6.39
*DDM1*	ATP-dependent DNA helicase DDM1	0	0.49	0.86
*DME*	Transcriptional activator DEMETER	2.16	0.48	0.6
*DRM1*	DNA(cytosine-5)-methyltransferase DRM1	2.04	0.47	0.35
*DRM2*	DNA(cytosine-5)-methyltransferase DRM2	0.72	0.47	0.72
*FAAH*	Fatty acid amide hydrolase	2.81	0.09	0.12
*FRU*	Acid beta-fructofuranosidase	5.59	0.19	0.14
*GA2ox6*	Gibberellin 2-oxidase	0.03	104.5	100
*GLU*	Beta-glucosidase bogh3b-like	3.34	0.18	0.12
*IAA32*	Auxin-responsive protein IAA32	4.49	3.32	1.52
*ILR1*	IAA-amino acid hydrolase ILR1	0.67	0.1	0.48
*LIM2*	LIM domain-containing protein WLIM2b	5.88	0.04	0.03
*LSD*	Lysine-specific demethylase	2.02	0.49	0.33
*MADS2*	MADS-box transcription factor 2	6.38	0.45	0.49
*MADS23*	MADS-box transcription factor 23	0.63	0.45	0.25
*AGL29*	MADS-box gene, AGL29	0.81	0.37	0.2
*AGL30*	MADS-box gene, AGL30	0.84	0.19	0.13
*AGL61*	MADS-box gene, AGL61	0.79	0/15.6	0.03
*AGL62*	MADS-box gene, AGL62	6.78	0.42	0.17
*AGL104*	MADS-box gene, AGL104	0.54	0.29	0.27
*MET1*	DNA methylation 1	1.46	0.5	0.58
*MYB21*	Myb21	0/2.1	14.08	2.81
*MYB26*	Myb26	0/1	0.5	0.47
*PHY*	Phytochrome B	0.58	0.46	0.33
*PK*	Pyruvate kinase, cytosolic isozyme	1.53	0.12	0.16
*ROS1*	ROS1, Repressor of silencing 1	1.34	0.46	0.3

Note: 0/2.1 means that the DNA methylation of P3A and P3B was 0 and 2.1, respectively.

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
