# Peer review of "Integrated Methylome and Transcriptome Analysis Provides Insights into the DNA Methylation Underlying the Mechanism of Cytoplasmic Male Sterility in Kenaf (Hibiscus cannabinus L.)"

_ijms, 2022, doi:10.3390/ijms23126864_

Round 1
Reviewer 1 Report
The paper “Integrated methylome and transcriptome analysis provides insights into the DNA methylation underlying the mechanism of cytoplasmic male sterility in kenaf (Hibiscus cannabinus L.) by Zengqiang Li et al. analyzed the DNA methylome and transcriptome in line with cytoplasmic male sterility (P3A) and its maintainer line (P3B) and found that several genes are characterized as both differentially methylated genes and differentially expressed genes (DEGs). And they concluded that DEGs were involved in carbon metabolism, plant hormone signal transduction, TCA cycle, and MAPK signaling pathway are important in the occurrence of CMS in kenaf.
The data may be useful and important. However, I don’t think that difference in DEGs and DMGs is the cause of CMS, it may be the result of CMS or other reason. Therefore, I think the author revise the title and the text.
In addition, I have several questions and comments.
- Figure 1. In each graph, the bars with a and b are same value (no significant difference?)
If each a and b indicate different value, it is better to use asterisk or something to show the significant difference.
- Fig. 3 The number under the transverse axis is nucleic acid number (bp)? If so, please indicate htat.
3.Fig. 3. L134-135 the authors state that DNA methylation level in kenaf male sterile line P3A was lower than that of its maintainer line P3B. However, I cannot understand that. It looks same. We need to know how to see the figure 3.
- Figure 8. there should be explanation of the figure. And explanation of color of the dot is required.
- In discussion part, the authors should indicate the figure or table number when they out mention of their results. for example, L280-285: although they are explaining their results, there is no indication of Figures or Table which they are talking.
- Figure 9. As mentioned above, I cannot agree with the model. In addition, because there is no explanation of the arrow head (in red) and dotted line, I cannot understand the model.
Minor point.
L115 and 116 KB to kb
Author Response
- Figure 1. In each graph, the bars with a and b are same value (no significant difference)? If each a and b indicate different value, it is better to use asterisk or something to show the significant difference.
Response: Thank you for your reminder. The different lowercase letters indicate significant differences at the P < 0.05 level, and we have added this to the figure note.
- Fig. 3 The number under the transverse axis is nucleic acid number (bp)? If so, please indicate that.
Response: In Fig. 3a, b, e, and f, the transverse axis represents the location of the methylation region at the TSS (or TTS) of genes, where a negative value indicates the number of bases upstream and a positive value indicates the number of bases downstream. In Fig. 3c and d, the transverse axis represents the degree of methylation, expressed as 0-1. We have added this text to the figure note.
- Fig. 3. L134-135 the authors state that DNA methylation level in kenaf male sterile line P3A was lower than that of its maintainer line P3B. However, I cannot understand that. It looks same. We need to know how to see the figure 3.
Response: The data on the distribution of methylation sites in the gene body indicate that the DNA methylation level in the kenaf male sterile line P3A was lower than that in its maintainer line P3B. Due to the large amount of data and the small difference between P3A and P3B, it is not easy to see in Fig. 3. In this revision, we provided data on the distribution of methylation sites in the gene body in Table S1.
- Figure 8. there should be explanation of the figure. And explanation of color of the dot is required.
Response: Red dots indicate that the methylation level of the gene is negatively correlated with its expression level, and blue dots indicate that the methylation level of the gene is positively correlated with its expression level. The grey shadow indicates that the correlation between the methylation level and expression level is not significant (P>0.05). We have added this text to the figure note.
- In discussion part, the authors should indicate the figure or table number when they out mention of their results. for example, L280-285: although they are explaining their results, there is no indication of Figures or Table which they are talking.
Response: We have indicated the figure or table number.
- Figure 9. As mentioned above, I cannot agree with the model. In addition, because there is no explanation of the arrow head (in red) and dotted line, I cannot understand the model.
Response: Thank you very much for your patient review and guidance. The upwards red arrowhead indicates that the content of this indicator is significantly increased in P3A, and the downwards red arrowhead indicates that the content of this indicator is significantly decreased in P3A. The solid line indicates that there has been sufficient and in-depth research, while the dotted line indicates that further in-depth research and verification are needed. We have added this text to the figure note.

Reviewer 2 Report
Thanks for the opportunity to review this research: "Integrated methylome and transcriptome analysis provides insights into the DNA methylation underlying the mechanism of cytoplasmic male sterility in kenaf (Hibiscus cannabinus L.)". The paper is good.
The authors must correct the spaces in lines 223 and 224
Check the year of references 19, 20 and 35
Author Response
The authors must correct the spaces in lines 223 and 224
Check the year of references 19, 20 and 35
Response: Thank you very much for your patient review and approval. We have checked and corrected the text.
Reviewer 3 Report
Dear Authors,
I have an opportunity to review manuscript entitled “Integrated methylome and transcriptome analysis provides insights into the DNA methylation underlying the mechanism of cytoplasmic male sterility in kenaf (Hibiscus cannabinus L.)” submitted to IJMS MDPI.
Authors using the anthers of a kenaf CMS line (P3A) and its maintainer line (P3B), a comparative physiological and DNA methylome combined with transcriptome analyses investigated.
Extensive analyses indicated that DNA methylome analysis revealed 650 differentially methylated genes (DMGs) with 313 up- and 337 down methylated. Moreover, transcriptome analysis characterized 1,788 differentially expressed genes (DEGs) with 558 up- and 1,230 downregulated genes in P3A compared with P3B. Furthermore, obtained results indicated that DEGs can be involved in carbon metabolism, plant hormone signal transduction, TCA cycle, and MAPK signaling pathway were shown to be important in the occurrence of cytoplasmic male sterility-CMS in kenaf.
Introduction was created as a sufficient background for the reader.
Materials and methods are clearly presented in a repetitive way.
Research are generally well designed, but some results should be present in a more clear, transparent way. For example Figures 1 and 2 are very difficult to read, please, correct that;
How to interpret the meaning of Figure 6- please describe it more transparent;
Why actin in discussion part is in almost all places in capital letter ?
Discussion is clear described – it is not easy to summarize so many datas, therefore the scheme in summary part was a very good choice;
Authors underlined, that “The results provide new thoughts and insights into the epigenetic mechanism of kenaf CMS and pollen development as well as a basis for understanding this complicated mechanism in other species” so please try to outline the future prospects coming from obtained results as well as further studies important in that subject.
Author Response
- Research are generally well designed, but some results should be present in a more clear, transparent way. For example Figures 1 and 2 are very difficult to read, please, correct that;
Response: Thank you very much for your patient review and guidance. We have made careful modifications.
- How to interpret the meaning of Figure 6- please describe it more transparent;
Response: Thank you for your suggestion. We have described it more clearly.
- Why actin in discussion part is in almost all places in capital letter?
Response: Thank you for your reminder. We have changed all instances to lowercase letters.
- Discussion is clear described – it is not easy to summarize so many datas, therefore the scheme in summary part was a very good choice;Authors underlined, that “The results provide new thoughts and insights into the epigenetic mechanism of kenaf CMS and pollen development as well as a basis for understanding this complicated mechanism in other species” so please try to outline the future prospects coming from obtained results as well as further studies important in that subject.
Response: Thank you for your excellent comments. We have added the related contents. ‘Based on the results of this study, we will further explore the functions of both DEGs and DMGs that are closely related to the occurrence of CMS, such as AUX, SUS6, AGL30, MYB21, and RAS-related proteins, and the role of DNA methylation in their gene expression.’
